# Boost the resilience of protected areas to shocks by reducing their dependency on tourism

F. Ollier D. Andrianambinina[1]☯*, Derek Schuurman[2]☯, Mamy A. Rakotoarijaona[1]†, Chantal N. Razanajovy[1], Honorath M. Ramparany[1], Serge C. Rafanoharana[3], H. Andry Rasamuel[3], Kevin D. Faragher[4], Patrick O. Waeber[5,6], Lucienne Wilmé[3]*

1 Madagascar National Parks, Ambatobe, Antananarivo, Madagascar, 2 London, England, 3 Madagascar Program, World Resources Institute Africa, Antananarivo, Madagascar, 4 World Resources Institute, Washington, DC, United States of America, 5 International Forest Management, Bern University of Applied Sciences, Bern, Switzerland, 6 Department of Environmental Systems Science, Forest Management and Development, Institute of Terrestrial Ecosystems, ETH Zürich, Zürich, Switzerland

☯ These authors contributed equally to this work.
† Deceased.
* ollier_cdcsi@mnparks.mg (FODA); lucienne.wilme@wri.org (LW)

**Data Availability Statement:** Data in Supporting Information.

## Abstract

Ecotourism is widely considered a strong mechanism for the sustainable funding of protected areas (PAs). Implemented during the 1990s in Madagascar, nature-based tourism experienced positive growth over the last 30 years with increasing numbers of visits to the parks and reserves. Revenue earned from entrance fees to the network of PAs managed by Madagascar National Parks has never been sufficient to finance their management. Political crises and the COVID-19 pandemic in particular, have highlighted for park managers, the risk of relying on such earnings when they covered just 1% of the required funding in 2021. Alternative mechanisms of funding are analysed for all of Madagascar's PAs with a view to facilitating sustainable conservation of the localities and protection of the island's biodiversity.

## Introduction

Mainland Africa is renowned for its trans-boundary peace parks and a host of other protected areas. Millions of tourists visit the classic African safari destinations such as Kenya, Tanzania, Botswana, Zambia, and South Africa, to observe larger mammals, especially the so-called 'Big Five' (African elephant, Cape buffalo, African lion, Leopard and Rhinoceros). While the dark spectre of wildlife crime perpetually looms in the background, threatening to undermine ecotourism (Box 1), sub-Saharan Africa's safari industry is estimated to have generated US$ 12.4 billion of annual revenues in 2019 [1] while the continent's wildlife and nature-based tourism industry as a whole, is estimated to generate in excess of US$ 29 billion annually [2].

Apart from safari tourism, wildlife watching tourism in Africa includes products such as Gorilla and Chimpanzee tracking in Central Africa and also on the theme of primate watching experiences, just across the Mozambique Channel, Madagascar offers Lemur watching as its

**Funding:** The authors received no specific funding for this work.

**Competing interests:** The authors have declared that no competing interests exist

## Box 1. Nature-based tourism and protected areas

Nature-based tourism is a broad term that encompasses any type of tourism that takes place in natural areas. It includes activities such as wildlife viewing, outdoor recreation, and nature-based activities, such as camping and beach vacations. While nature-based tourism can have some environmental and economic benefits, it is not necessarily sustainable or focused on conservation. Nature-based tourism thus allows people to connect with and benefit from ecosystems in various ways [3–7]. An increase in nature-based tourism can significantly impact the management of protected areas (PAs)—financial budgeting, infrastructure development, educational and tourist programmes, visitor management [8–11]. The Convention on Biological Diversity (CBD) emphasizes the importance of evaluating and sustainably managing nature-based tourism to reduce poverty and to protect the environment [12].

Ecotourism, considered a segment of nature-based tourism, is a type of tourism that aims to minimize the negative impacts on the environment and culture of host communities while providing educational and authentic experiences for tourists. It is a responsibly conducted form of tourism that aims to promote conservation and sustainability while also providing economic benefits to local communities [13, 14]. Additional forms of alternative tourism include green, responsible and good tourism [13] as well as conscious tourism where focus is on the development of more mindful, discerning travelers and on shifting focus from product to the development of meaningful experiences [15].

Protected areas (PAs) often serve as the foundation for nature-based tourism, as they provide opportunities for people to experience and learn about the natural world while also helping to conserve and protect it [16]. Studies have revealed that nature-based tourism tends to bring more visitors to those PAs with the highest levels of biodiversity [17–19]; to those which have been established longer [20]; those that are larger in size [21, 22], and those which are more readily accessible from urban areas [6]. Other factors which have been shown to influence visitor numbers include climate and weather [23, 24] as well as elevation [6]. It has been found that fewer people visit the more remote PAs [21], while PAs in high income countries tend to receive more visitors [25, 26].

For a long time, nature-based tourism conducted in a responsible manner and increasingly with sustainability in mind, has been regarded as a pivotal mechanism which contributes to the successful conservation of protected areas (PAs) by increasing their visibility and in so doing, attracting political attention, encouraging financial support, raising awareness of nature and ultimately, safeguarding biodiversity [27].

primary unique selling point. While Madagascar lacks the 'Big Five' or great apes, the island does have an exceptionally high degree of endemism in its biodiversity, including the >100 species of lemurs which only occur there. These charismatic non-human primates range in size from the monogamous, blue-eyed and ape-like Indri (*Indri indri*) which weighs in at some 7 kg, to the diminutive Madame Berthe's mouse-lemur (*Microcebus berthae*) which tips the scale at only 0.050 kg. The contrasting and fluffy coats of many of the larger lemur species, add to their appeal and popularity [28]. Of the world's eight baobab species, six occur only in Madagascar [29].

While Madagascar is less than a sixth the size of the Congo basin in Central Africa, it has more plant species than the latter (14 thousand VS. 10 thousand) and the rate of endemism is 80% for Madagascar plants as compared to 30% for the Congo basin. In Madagascar, rates of endemism among its faunal groups vary from 50% for the birds, to almost 100% for reptiles and amphibians [30]. With about 5000 km of shoreline, the island is mostly pantropical and contains a diversity of forests from humid and sub-humid in the East and Northeast, to dry deciduous and dry 'spiny bush' or sub-arid thorn thickets in the West and Southwest [30–32]. Throughout its coastal waters, Madagascar provides breeding and calving habitat for hump-back whales (*Megaptera novaeangliae*) from June to September before migrating back to polar waters where they feed [33, 34].

The island's terrestrial biodiversity is mostly concentrated in forests and to a lesser extent in wetlands, with a sprinkling of endemic taxa inhabiting the open landscapes dominated by grasslands [30, 32, 35, 36]. The largest and most undisturbed forests have been protected as Strict Nature Reserves since the first half of the 20th century. The protected areas network was created to safeguard maximal biodiversity and in so doing, cover most of the island's ecosystems, including the marine environment. A principle aspect has always been to ensure connectivity through corridors. Of the 123 PAs, 96 are terrestrial; 9 marine and 18 include both terrestrial and marine ecosystems. Nine of the eleven Strict Nature Reserves (IUCN Ia) have been turned into National Parks (IUCN II) to accommodate nature-based tourism. Forty-three PAs, with IUCN categories I, II, and IV [37], are managed by MNP—Madagascar National Parks, a parastatal organization.

The 1990s saw the emergence of the ecotourism concept after the World Bank and International Development Bank—motivated by global rainforest losses—had discontinued loans to mass tourism organizations, reinstating them in 1990 under ecotourism banner [38]. It was also during that period when ecotourism began to develop in Madagascar [39]. Annual numbers of inbound visitors to Madagascar increased slowly from approximately 100,000 in the mid-1990s to almost 500,000 in 2019, and this resulted in a total revenue just short of US\$ 1 billion [40]. Clearly therefore, tourism assumed a significant position within the total market value of final goods and services produced in Madagascar, i.e., around 6% Gross National Product (GNP) in 2019. While the statistics appear encouraging, it should be pointed out that before Madagascar's PA network was expanded in mid-2010, the collective earnings from tourism-derived entrance fees were significantly lower than the maintenance costs for effective PA management were [41]—and that this is still the case.

The onset of the COVID-19 pandemic brought the world to a quasi-halt in March 2020. The international aviation industry, which had registered almost 40 million flights in 2019 globally, recorded just below 17 million and 19 million flights for 2020 and 2021, respectively [42]. Tourism is one of the sectors that suffered most extensively as a result of the pandemic. In March 2020, Madagascar closed its borders, only reopening them again in April 2022. The protected areas were off limits to visitors from March to July 2020.

The objective of this study is to quantify the contribution of nature-based tourism to PA management in Madagascar. Our focus is on PAs that have higher levels of biodiversity, and which are managed for tourism, allowing for comparisons over different time periods. These PAs are regarded as being more appealing to visitors.

## Methodology

For this study we selected terrestrial protected areas (PAs) with IUCN Ia (Strict Nature Reserve); IUCN II (National Parks) managed for ecosystem protection and recreation and IUCN IV status (Habitat / Species Management Areas, managed for conservation through

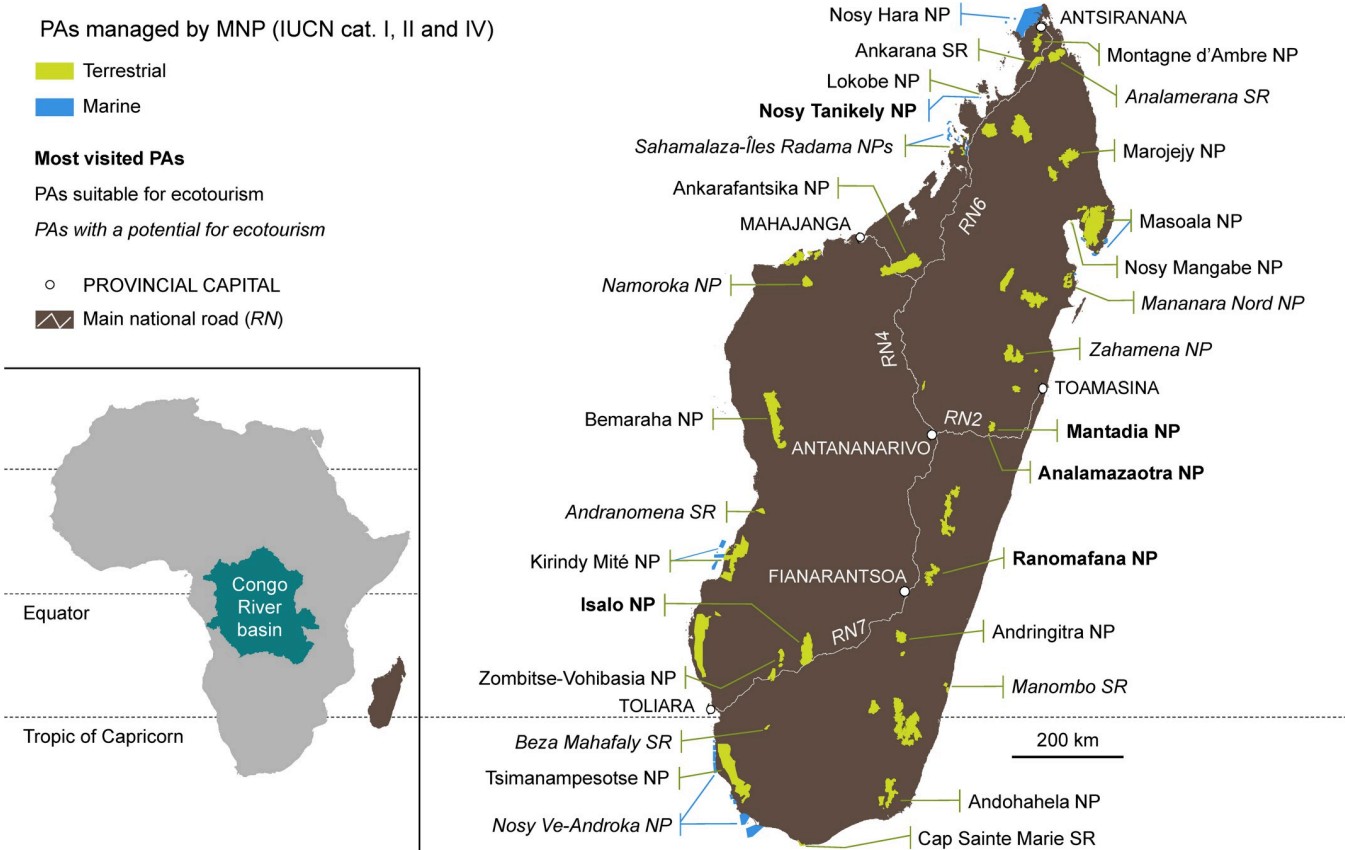

**Fig 1. Protected areas of Madagascar.** Protected areas in IUCN categories I, II and IV managed by Madagascar National Parks (MNP) and their importance for the implementation of ecotourism.

management interventions [43] (Fig 1). We have not included the other IUCN categories—firstly, IUCN III (Natural Monuments, managed for special natural features), of which only two examples, both small, exist—nor have we examined IUCN V and VI PAs as those are created for multiple usage and also, because they were set aside as recently as the mid-2010s [41, 44], they have a limited history.

Another filter we have applied in the selection of PAs is the management structure. We consider only PAs managed by MNP, because the centralized and standardized governance approach applied to them also facilitates access to data. We examined the age of the PAs, the changes in their IUCN categories over time, their size, and their accessibility, in order to understand each locality's prospects for tourism. The PAs considered, account for 43 parks and reserves managed by MNP: in 20, nature-based tourism has been implemented, in 10 there is potential for the development of this activity while the remaining 13, are not considered suitable for the development of tourism-related pursuits (Table A in S1 File).

We collected and collated annual and monthly visitor data spanning the 30-years period from 1992 to 2021. These data reveal the MNP income for PA management from tourism. To assess the net contribution of tourism to PA management, we also compiled and collated MNP's costs of managing the network of the PAs which they oversee, taking into account elements such as staff, external services, consumables and infrastructure / equipment, as well as their sources of funding from 2017–2021. We focused on these years because of amendments to the entrance fee structure at the end of 2016. We gathered the total number of tickets sold

for each Protected Area from January 1992 to December 2021 (Table B in S1 File). Note that the number of tickets sold does not reflect the number of individual visitors: there is no specific ID associated with any particular ticket, so an individual visitor can have several PA entries for different dates. All data are from MNP.

To ascertain the impact of the COVID-19 pandemic on the revenue generated by park entrance fees, we calculated a trend line for PA visits by focusing on the number of tickets sold during years that were not affected by major crises, i.e., the periods from 1992 to 2001; 2003 to 2009, 2011 to 2019 for terrestrial PAs, as well as the years 2011 to 2019 for marine PAs. The ten years from 1992 to 2001 were politically stable, while the year 2002 was marked by a political crisis in January following the presidential election during which Marc Ravalomanana opposed Didier Ratsiraka. The years 2009 and 2010 were strongly impacted by the January 2009 coup d'état after presidential elections, when Marc Ravalomanana and Andry Rajoelina were in opposition to one-another, and which resulted in prolonged political instability. In the process of calculating the increase in the numbers of PA visits since 1992, we consider 2011 as being a 'normal' year, even though it followed two political crises, a period of insecurity and sporadic failures in the provision of services to visitors. We estimated losses due to shortfalls in sales of entrance permits for 2020 and 2021 by examining trends calculated over the mentioned 30 years, and by examining the average distribution of visits per PA for the period 2010–2019. We scrutinized age and nationality of visitors after the establishment of the new entrance fees in 2016 in order to best estimate the losses incurred during the COVID-19 years (Table C in S1 File).

For data interpretation and validation, we conducted a survey among park managers of the main PAs managed by MNP who participated in a workshop in Antananarivo during February 2022. The 17 park managers were questioned about how they would explain a shortage or a change in tourist numbers to their parks and they were asked to elaborate on the socio-economic and/or political context pertaining to specific years in their regions.

While investigating alternative funding sources for park management, we also consulted with FAPBM (*Fondation pour les Aires Protégées et la Biodiversité de Madagascar*, Foundation for the protected areas and the biodiversity of Madagascar, see Box A in S1 File) to shed light on the mechanisms and elucidate the contribution of funds towards protected area management and conservation in Madagascar.

## Results

The number of tickets sold to enter parks and reserves from 1992 to 2019 increases over the years, with significant declines for certain years being followed by gradual recoveries (Fig 2). Considering the number of tickets sold for years not affected by major crises, i.e., the periods from 1992 to 2001; 2003 to 2009 and 2011 to 2019 for terrestrial PAs, as well as the years 2011 to 2019 for marine PAs, the increase in the number of tickets sold to enter parks and reserves is linear (Fig 2).

More than half (55.3%) of the tickets sold over the 30 years period were for four PAs, namely Isalo, Analamazaotra, Mantadia and Ranomafana NPs. The level of endemism in Isalo NP [45] is low while it is high in the other three NPs. When considering the top ten PAs visited for the 30 years period, little variation appears in the list, with the exception of the popularity of the marine Nosy Tanikely NP which attracted the highest numbers of tourists and the inclusion of the nearby (terrestrial) Lokobe NP (Fig 1, Tables D and E in S1 File).

Out of the 43 PAs managed by MNP, a large proportion, 38 (84%), were established during the last century (1927–1997), and only 5 PAs (16%)—three of which are marine—were set aside in 2007, 2011 and 2015. The oldest PAs were gazetted as Strict Nature Reserve

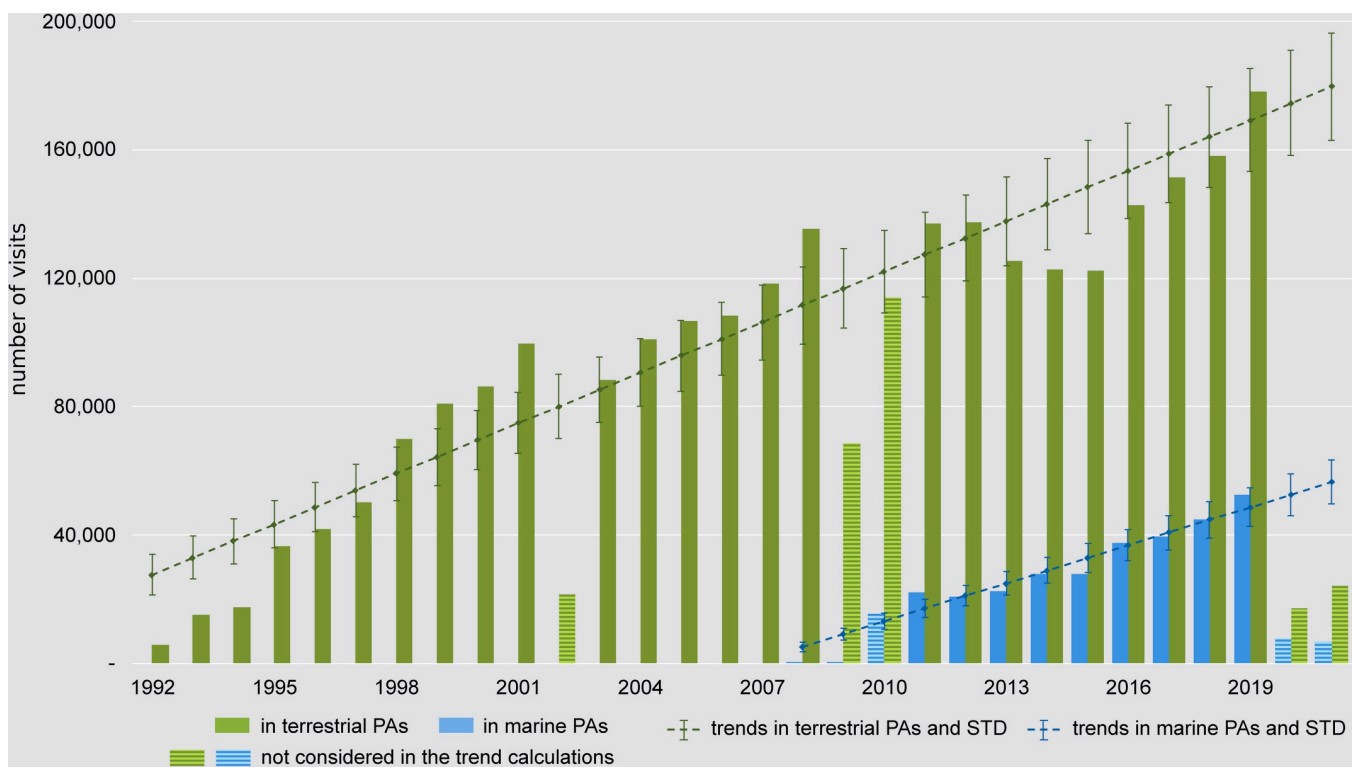

**Fig 2. Park entries.** Total number of visits in parks and reserves managed by MNP and trend lines calculated for both terrestrial protected areas and marine protected areas. Trend line for terrestrial PAs: number of visits = $5,246 * (year—1991) + 22,354$ ($r^2 = 0.90$); for marine PAs: number of visits = $3,952 * (year—2010) + 13,090$ ($r^2 = 0.93$); 2002, 2009–2010, and 2020–2021 are not considered in the estimation of the trends as they are years of political instability and in the case of 2020 and 2021 the COVID-19 pandemic; despite the financial crisis end of 2008, the year 2008 is regarded as a "normal" considering that visitors to Madagascar plan their travel well ahead of time, and number of visitors to PAs has not declined by the end of 2008).

(equivalent of IUCN cat. Ia) but 9 out of 11 have had their status changed to National Parks (IUCN cat. II) during the late 1990s to allow for ecotourism. The number of visitors in those 9 PAs increased until 2003, remained relatively constant from 2003 to 2013 and increased again from 2013 to 2019. The number of visitors decreased in Andohahela NP (Strict Nature Reserve until 1997) after 2008 due to political insecurity in the southeastern region of the country. The number of visitors to Bemaraha, Ankarafantsika and Andringitra NPs represent 77% of the total number of visitors recorded for these 11 NPs from 1997 to 2019 (Fig 3). Visits to the other parks did not plateau between 2003 and 2013. In 2003, several of the more popular parks received a low number of visitors after the 2002 political fallout. This was especially evident in Isalo, Mantadia, Analamazaotra, Montagne d'Ambre, Masoala NPs, and Ankarana SR. For the period 2013–2017, Isalo and Ranomafana NPs experienced a further decrease in the number of visitors, because of strike action among local guides. In 2012, MNP implemented training for the local guides, provided by an external partner contracted by the Ministry of Tourism. Only guides who passed the required training and who were therefore in possession of a certificate, were permitted to guide tourists after 2012. However, guides who had not undergone the training, repudiated this measure. At a meeting among the various stakeholders in 2016, it was agreed that the Ministry of Tourism would be the only entity in a position to stipulate the working conditions for local guides, and the situation then reverted to "normal". This is reflected in the increase of the number of tickets sold after 2017. Montagne d'Ambre NP and Ankarana SR experienced similar decreases in the numbers of tickets sold, because of a lack of domestic flights to the northern region, and insecurity surrounding these PAs following bandit

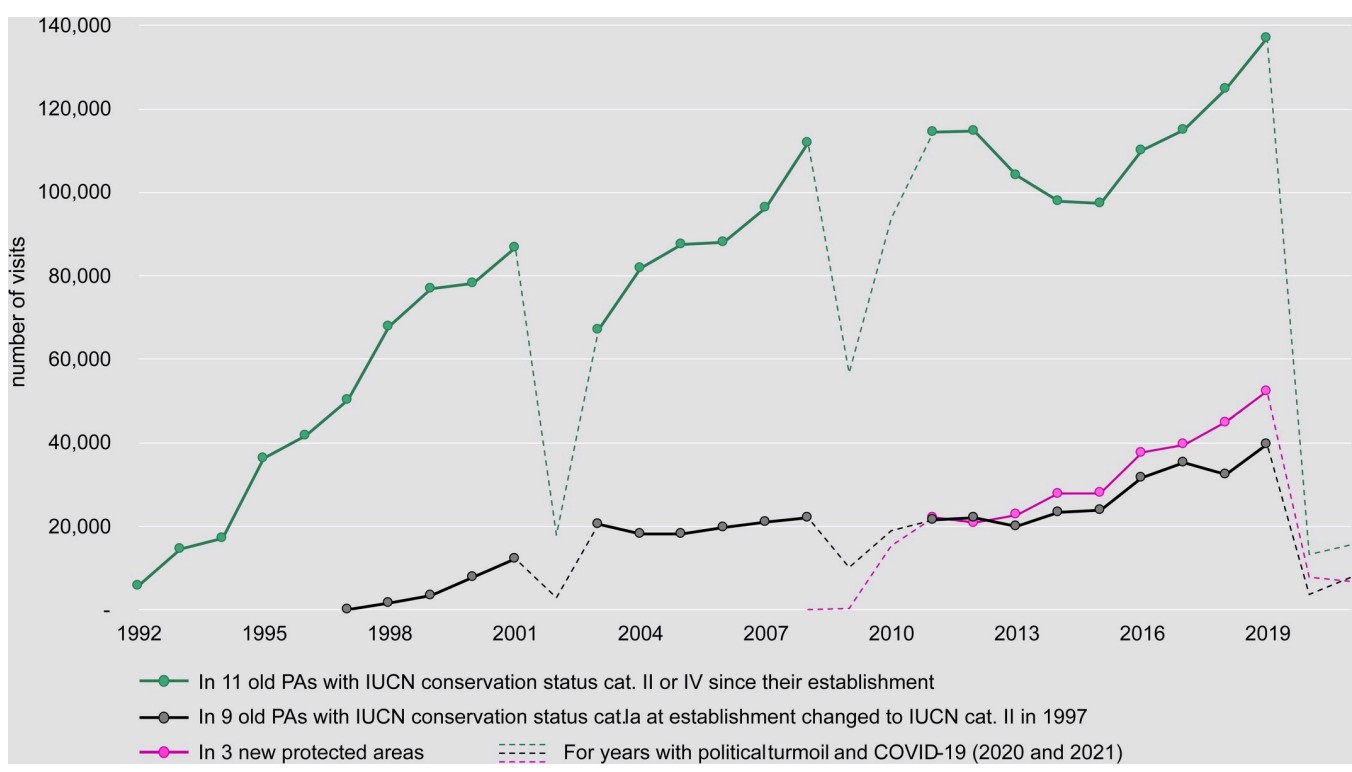

**Fig 3. Evolution of number of tickets sold for NPs which were Strict Nature Reserves before 1997, as compared to NP which were established as IUCN cat. II since their creation.** Years 2002, and 2009–2010 have been excluded from the main curve and appear on the dashed lines.

attacks on tourists during 2013. Visitor numbers started to increase again by 2017. Among the five new PAs managed by MNP, only Nosy Tanikely NP has attracted a large number of visitors: the tickets sold for this NP account for 97.7% of the tickets sold for the five new PAs. Ticket sales for the five new PAs accounted for only 14.0% of the total number of tickets sold in 2011. In 2019 the number of tickets sold for Nosy Tanikely NP alone, represented 22.0% of total ticket sales (Tables D and E in S1 File). Reasons for the markedly higher ticket sales to Nosy Tanikely include comparatively easier access; high visitor numbers to Nosy Be and lower entry fees for this PA (Table C in S1 File).

With the exception of the remaining two Strict Nature Reserves—which are entirely composed of a core area where no human activity other than research is permitted—all other PAs are comprised of one or more core area(s) surrounded by internal buffer areas in which a portion can be allocated to tourism and education [46, 47]. The latter usually accounts for 2.2% of the total area of the PAs, but can be less in smaller PAs such as Analamazaotra NP, which still attracts the highest number of visitors in proportion to surface area.

The most visited PAs are located in the western half of the island which traditionally, experiences a long dry season from April to October. Some exceptions include Mantadia, Analamazaotra and Ranomafana NPs, all situated in the humid eastern forest band, which experiences a lengthy rainy season. Elevation is not really a determining factor for tourism in Madagascar. It can be considered more as an opportunity for the promotion of some sites, such as Andringitra or Marojejy NPs which are recognized as high mountains.

From 2016 to 2019, the entrance fees and secondary or additional incomes from tourism covered only approximately 35–40% of the conservation management costs, compared to less than 20% during years 2013–2015. Over a 26-year period from 1995 to 2020, a total of

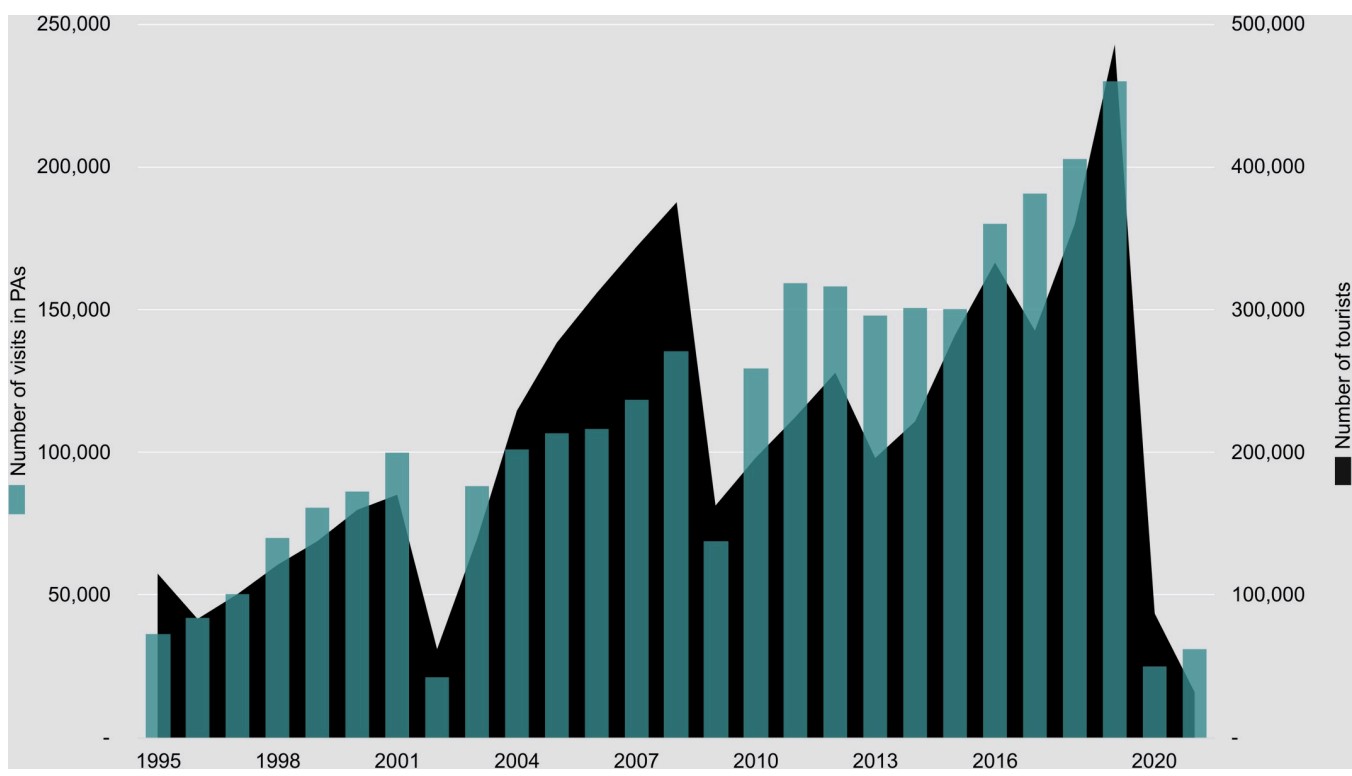

**Fig 4. Madagascar tourists and PA entry tickets.** Total number of tourists entering Madagascar VS. number of visits in parks and reserves under MNP management.

5,717,100 tourists was reported for Madagascar, generating an estimated revenue of US$ 10,5 billion for the period 1995–2020 [40, 48]. During this time, 2,938,736 tickets to enter PAs were sold (Fig 4).

The entrance fees for foreign tourists to PAs were increased in 2016 and these amended fees remained unchanged until December 2021. Most PAs adopted a rate of MGA 45,000 per day for foreigners (US$ 13.38 as of 2016), with higher rates applied to the most popular terrestrial PAs and lower rates for the popular, small marine Nosy Tanikely (Fig 1, Table C in S1 File).

The closing of borders and cessation of tourism during the COVID-19 pandemic had a more dramatic impact on numbers of entry tickets sold than any other crisis did. The major political events in 2002 and 2009—and their aftermaths—were also cited by park managers as reasons for noteworthy decreases of ticket sales. Other significant reasons for decreases in entry ticket sales include various failures on the part of the national airline which was unable to transport tourists to more remote destinations. After the 2009 coup d'état, prolonged insecurity and uncertainty led to a decrease in visitor numbers, especially to the Northern protected areas where safety and security issues also occurred. Compared with the total number of tourists in Madagascar, the total number of visits to PAs was proportionately less affected in the years following the 2002 political crisis than the years following the 2009 political fallout (Fig 4).

The FAPBM as a trust fund, is designed to provide grants to ensure the conservation of the protected areas and thereby, safeguarding of Madagascar's biodiversity. The initial capital of one million US dollars in 2005, was increased to US$ 139 million by the end of 2021. The interest generated annually is between 3 and 4%, however, the total capital remains at a deficit

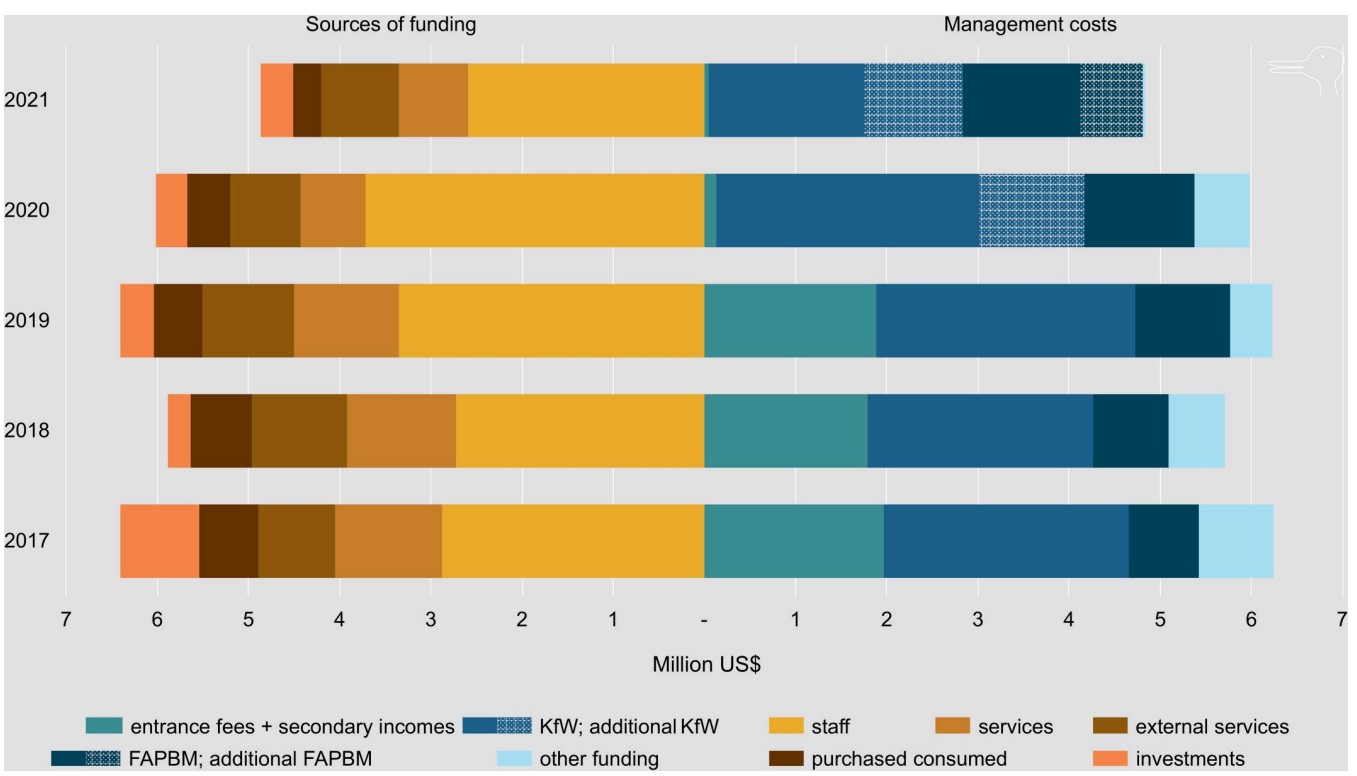

**Fig 5. PA funding Madagascar.** MNP funding for parks and reserves in categories I, II and IV—in million US$ dollar equivalents; the losses of entrance permit ticket sales for PAs managed by MNP have been estimated as being between US$ 3.7 and 5.9 million for 2020 and 2021—US$ 2,268,358—min = US$ 1,746,272; max = 2,855,348—in 2020 and US$ 2,415,991—min = US$ 1,878,324; max = 3,019,958—in 2021).

when balanced against costs required to implement and maintain the high number of conservation initiatives that are currently under way in Madagascar (Box A in S1 File).

Even in combination with additional income streams, entrance ticket sales have never generated enough revenue to cover the cost of the management of the PA network. This applies to the periods preceding and following the implementation of the increased rates in 2016 (Fig 5). MNP is predominantly supported by the German state-owned investment and development bank KfW and by FAPBM, both of which have been instrumental in dealing with the COVID-19 crisis. The emergency funds from FAPBM amount to US$ 100,000 per year, but this is inadequate to address the effects of crises such as COVID-19 (Box A in S1 File). Although a new COVID-19 fund has been implemented, MNP still relies heavily on finance from donors. To maintain the integrity of the PAs and to support the surrounding communities, KfW granted additional funds in 2020 and 2021. FAPBM also granted funds to compensate the communities living in the vicinity of PAs (Fig 5).

## Discussion

### The international perspective on tourism in Madagascar

Globally, natural capital is a significant contributor to the wealth of nations [49–51]. Tourism can and does play a significant role in conservation [52, 53], essentially linking intrinsic values surrounding amenity and biodiversity [54]. The pandemic has however revealed tourism to be a risky and a volatile sector that is vulnerable to external shocks. Financial sustainability understandably therefore, cannot be guaranteed for PAs if funding is to be solely based on revenue

generated from tourist visits. As we have outlined, PA entrance permit sales have never come close to covering the cost of conservation in Madagascar. This is the case despite strategic plans having been rolled out during the early implementation phase of the PA network in the 1990s and the National Environmental Action Plan during the 1990s [55].

The endemic biodiversity of Madagascar is characterized by a high level of micro-endemism with numerous, narrow-ranged species [56, 57]. This is sufficient reason to encourage multiple site visits whenever possible for tourists, who are therefore able to observe a wider range of biodiversity. The biodiversity—rich PAs include Mantadia, Analamazaotra, Ranomafana, Ankarana, Ankarafantsika and Masoala NPs—all of which are among the top 10 visited PAs. Interestingly, the number of tickets sold for Isalo NP outnumbers sales for any of the aforementioned six NPs. Isalo and Bemaraha NPs are geosites, and in the case of Isalo, access is straightforward, with the main RN7 road passing by the park. Additionally, a number of high caliber hotels provide comfortable accommodation.

The top 10 most visited PAs accounted for 92.5% of the entry tickets sold over the period 1992–2021, and for 91.5% of sales over the period 2011–2019. This is the case despite the inclusion of new PAs in the network and the change of status of certain PAs that previously, were not open to tourists. Political insecurity and remoteness are the most important factors influencing the number of visits in PAs. A national road leading towards any PA is clearly advantageous. High numbers of ticket sales for visits to the marine Nosy Tanikely NP and the terrestrial Lokobe NP on Nosy Be, demonstrate the allure of responsibly conducted or "conscious" seaside tourism [15]. Despite the number of PAs in the whole network, the selection of top 10 PAs visited might well remain as it currently is for years to come, though certain PAs, such as Nosy Hara NP, could potentially become another popular (marine) PA. Others like Kalambatritra NP, will most likely not attract visitors in the foreseeable future.

Through the implementation of the National Environmental Action Plan, the international donor community wanted Madagascar to create more protected areas. And the government of Madagascar was willing to expand its protected area network with financial support from the international donor community. However the promotion of PAs as the 'goose's golden eggs' turned out to be an ill-founded notion. As we have illustrated, overall, funding remains inadequate to cover the management costs of Madagascar's 123 PAs. The PA network includes many extremely remote sites. Nine Strict Nature Reserves have been reclassified as NPs, thus allowing for more development of tourism-related opportunities. Following a contract with resident populations who wish to safeguard the integrity of their park, it was agreed that tourism will not be implemented at Mikea NP [58, but see 59]. There are other remote PAs at which lack of infrastructure means that tourism will not be able to be promoted for decades to come. However, these more remote PAs still require funding. While entrance fees from ecotourism can benefit all the PAs in the network, it is clear that the financial solution lies elsewhere. This point is driven home when considering that even in the most successful years to date for tourism, revenue generated by entry permit sales has not even equaled a third of the funding required to manage the network of the PAs under MNP's management. Additionally, during the worst year to date (2021), the mentioned revenues dropped to a mere 1.0% of the necessary revenue.

Optimal achievement of the protection of natural ecosystems and of the benefits they provide, occurs when protected areas are supported by public policy. However, the management of protected areas often suffers because of budget deficits, especially in impoverished countries where governments do not—or cannot—allocate resources to cover required costs [60, 61]. In a recent study encapsulating almost a quarter of the world's protected areas, deficits of resources in staffing and budget were reported for more than three quarters of the PAs scrutinized [62]. Madagascar's PAs were not included among the sample evaluated, but had they

been, they would have increased the proportion of PAs receiving insufficient funding from the government.

## The local perspective on tourism in Madagascar

Madagascar's network of PAs hosts the majority of its endemic biodiversity and intact ecosystems [41], so the provision of ecosystem services [63–65], serves to promote human well-being both actively and passively. Tourism activities conducted within and near PAs are not strictly speaking, ecosystem services, but rather, benefits delivered by nature [65].

While MNP does not manage land outside PAs, it does work closely with communities living in the external buffer zones of the PAs. Before 2009, MNP's aim was to transfer half of the entrance fees generated by the parks to fund development projects benefiting the resident populations. The funds were allocated to Committees of Support to Protected Areas or COSAP, whose mission it was to identify micro-projects eligible for such funding. Villagers were grouped into associations, whose objective was to develop alternative activities for those creating pressure on the PAs. These associations would propose projects to the COSAP. Following the political crisis of 2009 and the subsequent loss of earnings from entry fees to the PAs, this redistribution was reduced, but MNP has continued to operate micro-development projects, albeit by means of utilising other sources of funding. Additionally, MNP has systematically been advocating the creation of Local Park Committees that bring together the communities living near the PAs. MNP actively hires and trains people from these communities. It facilitates the creation of diverse occupations such as guiding for ecotourism, as well as the performing of miscellaneous PA management tasks, including the organization of surveillance patrols [66].

It remains the intention of MNP to upgrade and maintain infrastructure in the more visited parks. Additionally, MNP has taken significant steps to improve the livelihoods of the communities residing near the parks: ongoing achievements include the building of schools and health facilities, and the developing of agricultural programmes providing viable alternatives for villagers to make their living outside of the protected areas without having to use resources originating from within PA boundaries. During the period of the National Environmental Action Plan [55], park entrance fees were equally shared with the local communities for development projects aimed at compensating people for losses of livelihood consequent to land reallocation for PAs—the process of which involved prohibition of access to the land and loss of the right to use the land [67].

As soon as a new PA is created, its managers are required to compensate the resident communities and to build capacity enabling development of the villages as well as establishing new sources of revenue. Before 2009, this was mainly achieved via the share of tourism-generated revenue, however currently, local development projects are executed with special funding from FAPBM. The aim here is to invest in the creation of activities and facilities which enable resident communities to obtain products formerly harvested inside the PA, to instead be harvested from outside of the PA perimeters. Following the 2009 political crisis and various attempts to share revenues evenly with the people living adjacent to PAs, MNP reorganized its support of the resident communities. Fair and equitable sharing of benefits became an increasingly challenging issue in a scenario where only a handful of PAs are in a position to be generating sufficient income to contribute towards other costs.

A significant portion of the money spent by tourists travelling to Madagascar 'leaks' out of the island, a common phenomenon observed in poor countries [68]. Leakage occurs when international organizations such as airlines, travel companies and hotels, pocket most of the profits. Hotels and restaurants in close proximity to the most visited PAs can earn well and

they do participate to some extent in the local economies. However they seldom invest in the public infrastructure required, for example that which facilitates access to the PAs. An independent review of the entire Madagascar-related tourism value and supply chains needs to be conducted, in order to reasonably increase the benefits for the communities residing around the PAs. An inclusion in such a review is scrutiny of mechanisms facilitating sharing of revenues and an analysis of ways to increase the participation of the resident community [69–71].

During the period from 1995 to 2021, the entrance fees in PAs managed by MNP were increased only once, at the end of 2016. With that increase of the ticket prices included, the total revenue generated from 1995 to 2021 by tourist visits to the MNP-managed PAs was less than US$ 20 million, i.e., lower than half of a percent of the revenues generated by tourism in general for Madagascar for these 27 years [32, 48].

## The need for finding sustainable alternatives

Madagascar's distinctive biodiversity is part of the global heritage. But who really pays for its conservation [72–74]? At this point in time, when most resources are being consumed by the wealthy 10% minority of the global population, it is patently unreasonable to expect that the often vastly poorer majority effectively has to pay for basic ecosystem services which benefit all of humanity. As the planet's 'lungs', forests are key contributors to health, stable climate, and the conservation of biodiversity [75–78]. Any economic activity should rightly therefore finance conservation—and in so doing, ensure our survival within equitable structures which benefit nature. It is appropriate that the international donor community covers part of the costs of the conservation of Madagascar's biodiversity and pledges its continued support in this regard. In order to achieve sustainability, nature-based tourism paired with grants from FABPM should finance the infrastructure development and bear the management costs for the entire PA network. The formation of a robust game plan geared to cater for future emergency situations is of paramount importance. Conservation governance will need alternative funding strategies if long-term financial sustainability is to be attained. With biodiversity conservation and sustainability being intertwined, we maintain that a constructive approach on the part of donors would be to consider investing funds in the PA and Biodiversity Foundation. In a world where current challenges include climate change, war, financial crises, political issues and COVID-19, the intensity of pressures on tropical forests is such that there is a real urgency for the creation and implementation of a financial support system which serves as a viable alternative to the current scenario of dependency on tourism. Only then, can harmonious balance be achieved for the sake of community well-being and biodiversity conservation.

## Supporting information

**S1 File.** Information, data and sources considered to document Madagascar's revenues from nature-based tourism and costs of management for conservation: The 43 Protected Areas considered in this study and their suitability for nature-based tourism (Table A). Tickets sold by MNP to visit PAs (terrestrial and marine) from 2017 to 2021 (Table B). Entrance fees rate for PAs not aligning to the rate of MGA 45, 000 starting in 2016 (Table C). Evolution of the top 10 PAs visited over 30 years and from 2011 to 2019 according to their main appeal (Table D). Top 10 PAs in number of tickets sold VS. Top 10 PAs in revenue generated from 2017 to 2019, according to their main appeal (Table E). Conservation Trust Funds: The Foundation for Protected Areas and Biodiversity of Madagascar (Fondation pour les Aires Protégées et la Biodiversité de Madagascar FAPBM) (Box A).
(PDF)

## Acknowledgments

We would like to acknowledge the PA managers for sharing their opinion and experiences related to the fluctuation of visitors in the PAs during the sanitary crisis, especially Mrs Juliette Randriamanarivo, Director of Ambohitantely SR, Mr Mandimby Heriniaina Andriambololona, Director Ankarafantsika NP, Mr Amidou Jaovita, Director Ankarana et Analamerana SRs, Mr Mamy Tsipakay Rakotobenandrasana, Director Tsingy de Bemaraha NP, Mr Diamondra Fananako Andriambololona, Director Andranomena SR and Kirindy Mité NP, Mr Zarasolo Gérard Bakarizafy, Director Lokobe NP, Mr Solo Hervé, Head of operational section Nosy Hara and Montagne d'Ambre NPs, Mr Mora Willy Covis, Director Anjanaharibe-Sud SR and Marojejy NP, Mr Jean Fidelis Rakotomanana, Director Nosy Mangabe and Masoala NPs, Mr Clerc Tsivolany, Head of operational section Ranomafana NP, Mr Nestor Rafenonirina, Director of Sahamalaza/Îles Radama NP, Mr Justin Rakotoarimanana. Director Beza Mahafaly, SR and Tsimanampesotse and Nosy Ve Androka NPs, Mr Hery Lala Ravelomanantsoa, Director Analamazaotra and Mantadia NPs, Mrs Landisoa Randimbison, Director Nosy Tanihely NP, Mr Benoit, Director Zahamena NP, Mrs Juliette Raharivololona, Director Zombitse-Vohibasia NP, Mr Andriahery Randriamalaza, Director Mikea NP and Mr Jocelyn Bezara, Director Baie de Baly and Tsingy de Namoroka NPs. We are also grateful to colleagues at MNP, especially Mrs Haingovolatiana Raobizo Rasamoela, Ecotourism and Nature Tours Officer, Ny Saina Randrianarifetra, Legal Affairs Officer, Liliane Parany, Projects and Follow-up infractions Officer and Ghislain Rakotoniaina, Management controller, for kindly participating in the data management related to the number of visitors and financial data related of the management. We are grateful to Niamh Brannigan for inspiring early discussions. We acknowledge the comments and information provided by FAPBM, and we would also like to thank the support of the project USAID Hay Tao in Madagascar.

## Author Contributions

**Conceptualization:** F. Ollier D. Andrianambinina, Derek Schuurman, Mamy A. Rakotoarijaona, Lucienne Wilmé.

**Data curation:** F. Ollier D. Andrianambinina, Chantal N. Razanajovy, Lucienne Wilmé.

**Formal analysis:** F. Ollier D. Andrianambinina, Lucienne Wilmé.

**Investigation:** F. Ollier D. Andrianambinina, Derek Schuurman, Patrick O. Waeber.

**Methodology:** F. Ollier D. Andrianambinina, Derek Schuurman, Lucienne Wilmé.

**Project administration:** F. Ollier D. Andrianambinina, Lucienne Wilmé.

**Resources:** F. Ollier D. Andrianambinina, Derek Schuurman, Chantal N. Razanajovy, Honorath M. Ramparany, Serge C. Rafanoharana, H. Andry Rasamuel, Kevin D. Faragher, Patrick O. Waeber, Lucienne Wilmé.

**Supervision:** F. Ollier D. Andrianambinina, Derek Schuurman, Patrick O. Waeber, Lucienne Wilmé.

**Validation:** F. Ollier D. Andrianambinina, Kevin D. Faragher, Lucienne Wilmé.

**Visualization:** Lucienne Wilmé.

**Writing – original draft:** F. Ollier D. Andrianambinina, Derek Schuurman, Honorath M. Ramparany, Serge C. Rafanoharana, Kevin D. Faragher, Patrick O. Waeber, Lucienne Wilmé.

**Writing – review & editing:** F. Ollier D. Andrianambinina, Derek Schuurman, Patrick O. Waeber, Lucienne Wilmé.

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
