## [Decision Letter · Decision Letter 0]

20 Dec 2022

PONE-D-22-31843Boost the resilience of Protected Areas to shocks by reducing their dependency on tourismPLOS ONE

Dear Dr. Andrianambinina,

Thank you for submitting your manuscript to PLOS ONE. After careful consideration, we feel that it has merit but does not fully meet PLOS ONE’s publication criteria as it currently stands. Therefore, we invite you to submit a revised version of the manuscript that addresses the points raised during the review process.

We look forward to receiving your revised manuscript.

Kind regards,

László Vasa, PhD

Academic Editor

PLOS ONE

Journal Requirements:

2. We note that Figure 1 in your submission contain [map/satellite] images which may be copyrighted. All PLOS content is published under the Creative Commons Attribution License (CC BY 4.0), which means that the manuscript, images, and Supporting Information files will be freely available online, and any third party is permitted to access, download, copy, distribute, and use these materials in any way, even commercially, with proper attribution. For these reasons, we cannot publish previously copyrighted maps or satellite images created using proprietary data, such as Google software (Google Maps, Street View, and Earth). For more information, see our copyright guidelines: http://journals.plos.org/plosone/s/licenses-and-copyright.

Reviewers' comments:

Reviewer's Responses to Questions

**Comments to the Author**

1. Is the manuscript technically sound, and do the data support the conclusions?

Reviewer #1: Yes

Reviewer #2: Yes

2. Has the statistical analysis been performed appropriately and rigorously? 

Reviewer #1: Yes

Reviewer #2: Yes

3. Have the authors made all data underlying the findings in their manuscript fully available?

Reviewer #1: Yes

Reviewer #2: Yes

4. Is the manuscript presented in an intelligible fashion and written in standard English?

Reviewer #1: Yes

Reviewer #2: No

5. Review Comments to the Author

Reviewer #1: The study deals with a current topic, both the investigated area and the topic are relevant from the point of view of the research. The structure of the study is logical, but I am missing the Literature review chapter, which summarizes relevant literature on the topic with a critical analysis. The objective of the investigation is not clear either, I recommend clarifying this. It is also recommended to present the correlations between the research questions and the applied methodology. The obtained results are novel.

Reviewer #2: Well written paper. I have only one comment. Authors wrote: "Nature-based or ecotourism is widely considered a strong mechanism for the sustainable funding of protected areas (PAs)."

Does the nature-based tourism equal to ecotourism? I think not. Please read and add these to the References.

https://www.tandfonline.com/doi/abs/10.1080/13032917.2007.9687211

https://journals.plos.org/plosone/article?id=10.1371/journal.pone.0250390

6. PLOS authors have the option to publish the peer review history of their article (what does this mean?). If published, this will include your full peer review and any attached files.

Reviewer #1: No

Reviewer #2: No

---

## [Author Response · Author response to Decision Letter 0]

29 Dec 2022

Our answers below as presented in our rebuttal

We formatted our contribution according to the form above

2. We note that Figure 1 in your submission contain [map/satellite] images which may be copyrighted. All PLOS content is published under the Creative Commons Attribution License (CC BY 4.0), which means that the manuscript, images, and Supporting Information files will be freely available online, and any third party is permitted to access, download, copy, distribute, and use these materials in any way, even commercially, with proper attribution. For these reasons, we cannot publish previously copyrighted maps or satellite images created using proprietary data, such as Google software (Google Maps, Street View, and Earth). For more information, see our copyright guidelines: http://journals.plos.org/plosone/s/licenses-and-copyright.

Please find enclosed the written permission from Madagascar National Parks, the copyright holder of the map produced, filled using the form http://journals.plos.org/plosone/s/file?id=7c09/content-permission-form.pdf

We have updated our reference list following comments by the reviewers

We have added the following references:

1. Daniel TC, Muhar A, Arnberger A, Aznar O, Boyd JW, Chan KM, Costanza R, Elmqvist T, Flint CG, Gobster PH, Grêt-Regamey A. Contributions of cultural services to the ecosystem services agenda. Proc Natl Acad Sci USA. 2012;109(23): 8812–9. https://doi.org/10.1073/pnas.111477310

2. Torland M, Weiler B, Moyle BD, Wolf ID. Are your ducks in a row? External and internal stakeholder perceptions of the benefits of parks in New South Wales, Australia. Manag Sport Leis. 2015;20(4): 211–37. https://doi.org/10.1080/23750472.2015.1028428

3. Wolf ID, Ainsworth GB, Crowley J. Transformative travel as a sustainable market niche for protected areas: a new development, marketing and conservation model. J Sustain Tour. 2017;25(11): 1650–73. https://doi.org/10.1080/09669582.2017.1302454

5. Nikolova M, Stoyanova V, Varadzhakova D, Ravnachka A. Cultural ecosystem services for development of nature-based tourism in Bulgaria. Journal of the Bulgarian Geographical Society. 2021;45: 81–7. https://doi.org/10.3897/jbgs.e78719

6. Alpízar F. The pricing of protected areas in nature-based tourism: A local perspective. Ecol Econ. 2006;56(2): 294-307. https://doi.org/10.1016/j.ecolecon.2005.02.005

7. Eagles PF. Research priorities in park tourism. J Sustain Tour. 2014;22(4): 528–49. https://doi.org/10.1080/09669582.2013.785554

8. Job H, Paesler F. Links between nature-based tourism, protected areas, poverty alleviation and crises—The example of Wasini Island (Kenya). J Outdoor Recreat Tour. 2013;1: 18–28. https://doi.org/10.1016/j.jort.2013.04.004

9. Kebete Y, Wondirad A. Visitor management and sustainable destination management nexus in Zegie Peninsula, Northern Ethiopia. J Dest Mark. 2019;13: 83–9. https://doi.org/10.1016/j.jdmm.2019.03.006

10. CBD (Convention of Biological Diversity). CBD/COP/DEC/XIII/3. 2016. https://www.cbd.int/decisions/cop/?m=cop-13

11. Mehmetoglu M. Accurately identifying and comparing sustainable tourists, nature-based tourists, and ecotourists on the basis of their environmental concerns. Int J Hosp Tour Adm. 2010;11(2): 171–99. https://doi.org/10.1080/15256481003732840

12. Ouboter DA, Kadosoe VS, Ouboter PE. Impact of ecotourism on abundance, diversity and activity patterns of medium-large terrestrial mammals at Brownsberg Nature Park, Suriname. PLoS ONE. 2021;16(6): e0250390. https://doi.org/10.1371/journal.pone.0250390

13. Pollock, A. Conscious travel: Signposts towards a new model for tourism. Contribution to the 2ndUNWTO Ethics and Tourism Congress Conscious Tourism for a New Era, September 12th, Quito. 2012. Available: https://skift.com/wp-content/uploads/2012/09/presentacion-anna-meira-pollock.pdf

14. Jamal T, Stronza A. Collaboration theory and tourism practice in protected areas: Stakeholders, structuring and sustainability. J Sustain Tour. 2009;17(2): 169–89. https://doi.org/10.1080/09669580802495741

45. Randrianaly HN, Di Cencio A, Rajaonarivo A, Raharimahefa T. A proposed geoheritage inventory system: case study of Isalo National Park, Madagascar. J Geosci Environ Prot. 2016;4(5): 163–72. https://doi.org/10.4236/gep.2016.45016

46. COAP. Repoblikan’i Madagasikara. Assemblée Nationale. Loi n° 2015-005 Refonte du Code de Gestion des Aires Protégées. 2015. Available: https://edbm.mg/wp-content/uploads/2017/12/Loi-n-2015-005_COAP.pdf

47. COAP. Repoblikan’i Madagasikara. Ministère de l’E,vironnement et du Développement Durable. Décret N° 2017-415 du 30 mai 2017 fixant les modalités et les conditions d’application de la LOI n° 2015- 005 du 26 février 2015 portant refonte du Code de Gestion des Aires Protégées. Available: https://www.environnement.mg/wp-content/uploads/2019/04/DECRET-COAP-Sign%C3%A9.pdf

56. Wilmé L, Goodman SM, Ganzhorn JU. Biogeographic evolution of Madagascar's microendemic biota. Science. 2006;312(5776): 1063–5. https://doi.org/10.1126/science.1122806

57. Wilmé L, Ravokatra M, Dolch R, Schuurman D, Mathieu E, Schuetz H, Waeber PO. Toponyms for centers of endemism in Madagascar. Madag Conserv Dev. 2012;7(1): 30–40. https://doi.org/10.4314/mcd.v7i1.6

66. Andrianambinina FOD, Waeber, PO, Schuurman, D, Lowry II, PP, Wilmé, L. Clarification on protected area management efforts in Madagascar during periods of heightened uncertainty and instability. Madag Conserv Dev. 2022;17(1): 25–28. https://dx.doi.org/10.4314/mcd.v17i1.7

We have not deleted any references from the previous manuscript.

Comments to the Author

4. Is the manuscript presented in an intelligible fashion and written in standard English?

Reviewer #1: Yes

Reviewer #2: No

This relates to the point 5. See below

5. Review Comments to the Author

Reviewer #1: The study deals with a current topic, both the investigated area and the topic are relevant from the point of view of the research. 

5a. The structure of the study is logical, but I am missing the Literature review chapter, which summarizes relevant literature on the topic with a critical analysis. 

We have provided a literature review in the form of Box1 in which we move part of our introduction and added a literature review. 

We have added the following:

“Nature-based tourism is a broad term that encompasses any type of tourism that takes place in natural areas. It includes activities such as wildlife viewing, outdoor recreation, and nature-based activities, such as camping and beach vacations. While nature-based tourism can have some environmental and economic benefits, it is not necessarily sustainable or focused on conservation. Nature-based tourism thus allows people to connect with and benefit from ecosystems in various ways[1–5]. An increase in nature-based tourism can significantly impact the management of protected areas (PAs)—financial budgeting, infrastructure development, educational and tourist programmes, visitor management[6–9]. The Convention on Biological Diversity (CBD) emphasizes the importance of evaluating and sustainably managing nature-based tourism to reduce poverty and to protect the environment[10].

Ecotourism, considered a segment of nature-based tourism, is a type of tourism that aims to minimize the negative impacts on the environment and culture of host communities while providing educational and authentic experiences for tourists. It is a responsibly conducted form of tourism that aims to promote conservation and sustainability while also providing economic benefits to local communities[11–12]. Additional forms of alternative tourism include green, responsible and good tourism[11] as well as conscious tourism where focus is on the development of more mindful, discerning travelers and on shifting focus from product to the development of meaningful experiences[13].

Protected areas (PAs) often serve as the foundation for nature-based tourism, as they provide opportunities for people to experience and learn about the natural world while also helping to conserve and protect it[14].

And move this part of the introduction at the end of Box1:

“Studies have revealed that nature-based tourism tends to bring more visitors to those PAs with the highest levels of biodiversity [15–17]; to those which have been established longer[18]; those that are larger in size [19,20], and those which are more readily accessible from urban areas[4]. Other factors which have been shown to influence visitor numbers include climate and weather[21,22] as well as elevation[4]. It has been found that fewer people visit the more remote PAs[19], while PAs in high income countries tend to receive more visitors [23,24]. 

For a long time, nature-based tourism conducted in a responsible manner and increasingly with sustainability in mind, has been regarded as a pivotal mechanism which contributes to the successful conservation of protected areas (PAs) by increasing their visibility and in so doing, attracting political attention, encouraging financial support, raising awareness of nature and ultimately, safeguarding biodiversity[25].”

In the discussion, we have added the following:

“The endemic biodiversity of Madagascar is characterized by a high level of micro-endemism with numerous, narrow-ranged species[56,57]. This is sufficient reason to encourage multiple site visits whenever possible for tourists, who are therefore able to observe a wider range of biodiversity. The biodiversity—rich PAs include Mantadia, Analamazaotra, Ranomafana, Ankarana, Ankarafantsika and Masoala NPs—all of which are among the top 10 visited PAs. Interestingly, the number of tickets sold for Isalo NP outnumbers sales for any of the aforementioned six NPs. Isalo and Bemaraha NPs are geosites, and in the case of Isalo, access is straightforward, with the main RN7 road passing by the park. Additionally, a number of high caliber hotels provide comfortable accommodation.

The top 10 most visited PAs accounted for 92.5 % of the entry tickets sold over the period 1992–2021, and for 91.5 % of sales over the period 2011–2019. This is the case despite the inclusion of new PAs in the network and the change of status of certain PAs that previously, were not open to tourists. Political insecurity and remoteness are the most important factors influencing the number of visits in PAs. A national road leading towards any PA is clearly advantageous. High numbers of ticket sales for visits to the marine Nosy Tanikely NP and the terrestrial Lokobe NP on Nosy Be, demonstrate the allure of responsibly conducted or “conscious” seaside tourism[13]. Despite the number of PAs in the whole network, the selection of top 10 PAs visited might well remain as it currently is for years to come, though certain PAs, such as Nosy Hara NP, could potentially become another popular (marine) PA. Others like Kalambatritra NP, will most likely not attract visitors in the foreseeable future.”

5b. The objective of the investigation is not clear either, I recommend clarifying this. 

We have changed the last § of the introduction as following:

“The objective of this study is to quantify the contribution of nature-based tourism to PA management costs in Madagascar. Our focus is on PAs that have higher levels of biodiversity, and which are managed for tourism, allowing for comparisons over different time periods. These PAs are regarded as being more appealing to visitors.”

5c. It is also recommended to present the correlations between the research questions and the applied methodology. The obtained results are novel.

In the methodology, we have added the following:

“We examined the age of the PAs, the changes in their IUCN categories over time, their size, and their accessibility, in order to understand each locality’s prospects for tourism.”

We have produced more analysis, added one figure (current fig 3) and the following text in the results, following our methodology:

“More than half (55.3 %) of the tickets sold over the 30 years period were for four PAs, namely Isalo, Analamazaotra, Mantadia and Ranomafana NPs. The level of endemism in Isalo NP[45] is low while it is high in the other three NPs. When considering the top ten PAs visited for the 30 years period, little variation appears in the list, with the exception of the popularity of the marine Nosy Tanikely NP which attracted the highest numbers of tourists and the inclusion of the nearby (terrestrial) Lokobe NP (Fig 1, Tables D and E in S1 file).

Out of the 43 PAs managed by MNP, a large proportion, 38 (84 %), were established during the last century (1927–1997), and only 5 PAs (16 %)—three of which are marine—were set aside in 2007, 2011 and 2015. The oldest PAs were gazetted as Strict Nature Reserve (equivalent of IUCN cat. Ia) but 9 out of 11 have had their status changed to National Parks (IUCN cat. II) during the late 1990s to allow for ecotourism. The number of visitors in those 9 PAs increased until 2003, remained relatively constant from 2003 to 2013 and increased again from 2013 to 2019. The number of visitors decreased in Andohahela NP (Strict Nature Reserve until 1997) after 2008 due to political insecurity in the southeastern region of the country. The number of visitors to Bemaraha, Ankarafantsika and Andringitra NPs represent 77 % of the total number of visitors recorded for these 11 NPs from 1997 to 2019 (Fig 3). Visits to the other parks did not plateau between 2003 and 2013. In 2003, several of the more popular parks received a low number of visitors after the 2002 political fallout. This was especially evident in Isalo, Mantadia, Analamazaotra, Montagne d’Ambre, Masoala NPs, and Ankarana SR. For the period 2013–2017, Isalo and Ranomafana NPs experienced a further decrease in the number of visitors, because of strike action among local guides. In 2012, MNP implemented training for the local guides, provided by an external partner contracted by the Ministry of Tourism. Only guides who passed the required training and who were therefore in possession of a certificate, were permitted to guide tourists after 2012. However, guides who had not undergone the training, repudiated this measure. At a meeting among the various stakeholders in 2016, it was agreed that the Ministry of Tourism would be the only entity in a position to stipulate the working conditions for local guides, and the situation then reverted to “normal”. This is reflected in the increase of the number of tickets sold after 2017. Montagne d’Ambre NP and Ankarana SR experienced similar decreases in the numbers of tickets sold, because of a lack of domestic flights to the northern region, and insecurity surrounding these PAs following bandit attacks on tourists during 2013. Visitor numbers started to increase again by 2017. Among the five new PAs managed by MNP, only Nosy Tanikely NP has attracted a large number of visitors: the tickets sold for this NP account for 97.7 % of the tickets sold for the five new PAs. Ticket sales for the five new PAs accounted for only 14.0 % of the total number of tickets sold in 2011. In 2019 the number of tickets sold for Nosy Tanikely NP alone, represented 22.0 % of total ticket sales (Tables D and E in S1 file). Reasons for the markedly higher ticket sales to Nosy Tanikely include comparatively easier access; high visitor numbers to Nosy Be and lower entry fees for this PA (Table C in S1 file).

With the exception of the remaining two Strict Nature Reserves—which are entirely composed of a core area where no human activity other than research is permitted—all other PAs are comprised of one or more core area(s) surrounded by internal buffer areas in which a portion can be allocated to tourism and education[46,47].The latter usually accounts for 2.2 % of the total area of the PAs, but can be less in smaller PAs such as Analamazaotra NP, which still attracts the highest number of visitors in proportion to surface area.

The most visited PAs are located in the western half of the island which traditionally, experiences a long dry season from April to October. Some exceptions include Mantadia, Analamazaotra and Ranomafana NPs, all situated in the humid eastern forest band, which experiences a lengthy rainy season. Elevation is not really a determining factor for tourism in Madagascar. It can be considered more as an opportunity for the promotion of some sites, such as Andringitra or Marojejy NPs which are recognized as high mountains.”

In the discussion, we have added the following:

“The endemic biodiversity of Madagascar is characterized by a high level of micro-endemism with numerous, narrow-ranged species[56,57]. This is sufficient reason to encourage multiple site visits whenever possible for tourists, who are therefore able to observe a wider range of biodiversity. The biodiversity—rich PAs include Mantadia, Analamazaotra, Ranomafana, Ankarana, Ankarafantsika and Masoala NPs—all of which are among the top 10 visited PAs. Interestingly, the number of tickets sold for Isalo NP outnumbers sales for any of the aforementioned six NPs. Isalo and Bemaraha NPs are geosites, and in the case of Isalo, access is straightforward, with the main RN7 road passing by the park. Additionally, a number of high caliber hotels provide comfortable accommodation.

The top 10 most visited PAs accounted for 92.5 % of the entry tickets sold over the period 1992–2021, and for 91.5 % of sales over the period 2011–2019. This is the case despite the inclusion of new PAs in the network and the change of status of certain PAs that previously, were not open to tourists. Political insecurity and remoteness are the most important factors influencing the number of visits in PAs. A national road leading towards any PA is clearly advantageous. High numbers of ticket sales for visits to the marine Nosy Tanikely NP and the terrestrial Lokobe NP on Nosy Be, demonstrate the allure of responsibly conducted or “conscious” seaside tourism[13]. Despite the number of PAs in the whole network, the selection of top 10 PAs visited might well remain as it currently is for years to come, though certain PAs, such as Nosy Hara NP, could potentially become another popular (marine) PA. Others like Kalambatritra NP, will most likely not attract visitors in the foreseeable future.”

Reviewer #2: Well written paper. I have only one comment. Authors wrote: "Nature-based or ecotourism is widely considered a strong mechanism for the sustainable funding of protected areas (PAs)."

Does the nature-based tourism equal to ecotourism? I think not. Please read and add these to the References.

https://www.tandfonline.com/doi/abs/10.1080/13032917.2007.9687211

https://journals.plos.org/plosone/article?id=10.1371/journal.pone.0250390

We have included details on ecotourism and nature-based tourism in the form of Box1 as following:

“Box 1. Nature-based tourism and protected areas. 

Nature-based tourism is a broad term that encompasses any type of tourism that takes place in natural areas. It includes activities such as wildlife viewing, outdoor recreation, and nature-based activities, such as camping and beach vacations. While nature-based tourism can have some environmental and economic benefits, it is not necessarily sustainable or focused on conservation. Nature-based tourism thus allows people to connect with and benefit from ecosystems in various ways[1–5]. An increase in nature-based tourism can significantly impact the management of protected areas (PAs)—financial budgeting, infrastructure development, educational and tourist programmes, visitor management[6–9]. The Convention on Biological Diversity (CBD) emphasizes the importance of evaluating and sustainably managing nature-based tourism to reduce poverty and to protect the environment[10].

Ecotourism, considered a segment of nature-based tourism, is a type of tourism that aims to minimize the negative impacts on the environment and culture of host communities while providing educational and authentic experiences for tourists. It is a responsibly conducted form of tourism that aims to promote conservation and sustainability while also providing economic benefits to local communities[11–12]. Additional forms of alternative tourism include green, responsible and good tourism[11] as well as conscious tourism where focus is on the development of more mindful, discerning travelers and on shifting focus from product to the development of meaningful experiences[13].

Protected areas (PAs) often serve as the foundation for nature-based tourism, as they provide opportunities for people to experience and learn about the natural world while also helping to conserve and protect it[14].”

Within the text, we were very careful in using ecotourism and nature-based tourism, or tourism accordingly.

---

## [Editor Report · Decision Letter 1]

27 Mar 2023

Boost the resilience of protected areas to shocks by reducing their dependency on tourism

PONE-D-22-31843R1

Dear Dr. Andrianambinina,

We’re pleased to inform you that your manuscript has been judged scientifically suitable for publication and will be formally accepted for publication once it meets all outstanding technical requirements.

Kind regards,

László Vasa, PhD

Academic Editor

PLOS ONE
---

## [Editor Report · Acceptance letter]

4 Apr 2023

PONE-D-22-31843R1 

Boost the resilience of protected areas to shocks by reducing their dependency on tourism 

Dear Dr. Andrianambinina:

I'm pleased to inform you that your manuscript has been deemed suitable for publication in PLOS ONE. Congratulations! Your manuscript is now with our production department. 

Kind regards, 

on behalf of

Prof. Dr. László Vasa 

Academic Editor

PLOS ONE